**Biogeosciences** Open Access

**Discussions**

# Rates and pathways of CH$_4$ oxidation in ferruginous Lake Matano, Indonesia

A. Sturm[1], D. A. Fowle[1], C. Jones[2,4], K. Leslie[1], S. Nomosatryo[3], C. Henny[3], D. E. Canfield[4], and S. A. Crowe[2,4]

[1]Department of Geology, University of Kansas, Lawrence KS 66047, USA
[2]Department of Microbiology and Immunology and Department of Earth, Ocean, and Atmospheric Sciences, University of British Columbia, 2350 Health Sciences Mall, Vancouver, British Columbia V6T 1Z3, Canada
[3]Research Center for Limnology, Indonesian Institute of Sciences (LIPI), Cibinong-Bogor, Indonesia
[4]Nordic Center for Earth Evolution, Institute of Biology, Univ. of Southern Denmark, Campusvej 55, 5230 Odense, Denmark

Received: 14 October 2015 – Accepted: 15 October 2015 – Published: 22 February 2016

Correspondence to: S. A. Crowe (sean.crowe@ubc.ca)

Published by Copernicus Publications on behalf of the European Geosciences Union.

## Abstract

This study evaluates rates and pathways of methane ($CH_4$) oxidation and uptake us-ing $^{14}$C-based tracer experiments throughout the oxic and anoxic waters of ferruginous Lake Matano. Methane oxidation rates in Lake Matano are low compared to other lakes,
but are sufficiently high to preclude strong $CH_4$ fluxes to the atmosphere. In addition to aerobic $CH_4$ oxidation, which takes place in Lake Matano's oxic mixolimnion, we also detected $CH_4$ oxidation in Lake Matano's anoxic ferruginous waters. Here, $CH_4$ oxidation proceeds in the apparent absence of oxygen ($O_2$) and instead appears to be coupled to nitrate ($NO_3^-$), nitrite ($NO_2^-$), iron (Fe), or manganese (Mn) reduction.
Throughout the lake, the fraction of $CH_4$ carbon that is assimilated vs. oxidized to car-bon dioxide ($CO_2$) is high, indicating extensive $CH_4$ conversion to biomass and under-scoring the importance of $CH_4$ as a carbon and energy source in Lake Matano and potentially other ferruginous or low productivity environments.

## 1   Introduction

Methane ($CH_4$) is a critical component of the global carbon cycle and is a potent green-house gas (Cicerone and Oremland, 1988; Conrad, 2009; Kroeger et al., 2011). In-deed, dramatic changes in global climate throughout Earth's history have been linked to alterations in the global $CH_4$ cycle (Kroeger et al., 2011; Sloan et al., 1992; Zeebe et al., 2009). Methane is produced in the environment as the end product of organic
matter degradation via methanogenesis, either through acetate fermentation or $CO_2$ reduction in freshwater and marine sediments and soils. A significant fraction of this $CH_4$ is then consumed through microbially catalyzed oxidation directly within soils and sediments or within water columns. Microbial oxidation of $CH_4$ can take place aerobi-cally, with $O_2$ as the electron acceptor, or anaerobically (AOM), typically with sulfate
($SO_4^{2-}$) as the electron acceptor (Boetius et al., 2000; Devol et al., 1984; Martens and Berner, 1974; Reeburgh, 1980). Indeed, sulfate-dependent anaerobic $CH_4$ oxidation

Discussion Paper | Discussion Paper | Discussion Paper | Discussion Paper |

**BGD**

doi:10.5194/bg-2015-533

**Rates and pathways of $CH_4$ oxidation**

A. Sturm et al.

Interactive Discussion

consumes most of the $CH_4$ produced in marine sediments (Devol et al., 1984; Knittel and Boetius, 2009; Martens and Berner, 1974; Reeburgh, 2007; Treude et al., 2005) and is therefore the largest natural sink for $CH_4$ on the planet (Reeburgh, 2007). The scarcity of $SO_4^{2-}$ in freshwater settings, however, is thought to largely preclude AOM, thus aerobic $CH_4$ oxidation is believed to dominate $CH_4$ consumption in these environments.

Large uncertainties accompany $CH_4$ budgets for freshwater environments due to physical and chemical diversity of lake, wetland, and soil systems, (Liikanen and Martikainen, 2003; Liikanen et al., 2003; Luesken et al., 2011a). Large intervals of diffusion limited marine sediments contain abundant sulfate and are therefore favorable to AOM. These represent a more effective $CH_4$ sink than the often narrow layers of oxic marine sediment that typically support aerobic methanotrophy. Sulfate concentrations are generally much lower in freshwater systems and, without expansive zones supporting sulfate-dependent AOM, may not be as effective at total CH4 removal as their marine counterparts. Thus, freshwater systems emit proportionally much more $CH_4$ to the atmosphere (Capone and Kiene, 1988; Conrad, 2009).

Recently, evidence has emerged for AOM utilizing alternative electron acceptors. Nitrate dependent AOM takes place in enrichment cultures from $NO_3^-$ and $NO_2^-$ rich canal waters (Raghoebarsing et al., 2006) and wastewater systems (Kampman et al., 2012; Luesken et al., 2011a, b; Shen and Hu, 2012). Isolates of *Methylomirabilis oxyfera* conduct nitrate-dependent $CH_4$ oxidation through a novel denitrifying pathway (Ettwig et al., 2010). Other $NO_3^-$-dependent $CH_4$ oxidizing archaea couple $NO_3^-$ reduction to ammonium ($NH_4^+$) with $CH_4$ oxidation (Haroon et al., 2013). Despite abundant documentation of $NO_3^-$ dependent AOM in lab settings and wastewaters, its broader significance in natural environments is largely unknown at this time. However, natural systems rich in $NO_3^-$ with rapid N-cycling, such as freshwater lakes, estuaries, and wetlands, should have the potential to oxidize immense amounts of $CH_4$ through $NO_3^-$ dependent AOM (Joye et al., 1999). Evidence for Fe and Mn dependent $CH_4$ oxidation is also accruing and has been proposed to explain geochemical evidence for AOM in

**BGD**

doi:10.5194/bg-2015-533

**Rates and pathways of CH$_4$ oxidation**

A. Sturm et al.

freshwater environments in the apparent absence of $SO_4^{2-}$ (Crowe et al., 2008a, 2011; Lopes et al., 2011; Nordi et al., 2013; Sivan et al., 2011). However, these environmental observations remain unsubstantiated, and AOM in these systems has not been confirmed by process rate measurements or shown through microbial isolation and pure culture laboratory experiments.

Lake Matano, Sulawesi Island, Indonesia, hosts the largest, deepest, and oldest known ferruginous basin on Earth (Crowe et al., 2008a, b). This system offers a unique opportunity to examine $CH_4$ biogeochemistry under conditions with very low natural $SO_4^{2-}$ concentrations and an abundant supply of Fe and Mn oxyhydroxides. In general, tropical lakes are understudied by comparison to their temperate counterparts. Further examination of such systems is needed to constrain global $CH_4$ budgets. Furthermore, with chemistry and physics analogous to those proposed for the Precambrianoceans, Lake Matano also affords opportunities to test models for the marine $CH_4$ cycle throughout much of Earth's history. To betterconstrain $CH_4$ biogeochemistry in Lake Matano, tropical lakes, and ferruginous environments in general, we have determined $CH_4$ oxidation rates with a suite of oxic and anoxic incubations using [14]C labeled $CH_4$. These rate measurements are interpreted with respect to the availability of oxidants to place constraints on the pathways of $CH_4$ oxidation throughout the lake.

## 2   Materials and methods

### 2.1   Sampling and general analyses

Samples were retrieved from a deep-water master station ($2°28'00''$ S and $121°17'00''$ E) (Crowe et al., 2008b) in May 2010. For deep and shallow water sampling (i.e. < 100 m or > 140 m) 5 L Go-Flow (Niskin; General Oceanics, Miami, FL, USA) bottles were used with a manual winch setup and a Furuno FCV585 sonar device for bottle placement by echolocation, achieving an accuracy and precision of ±1 m. For samples in intermediate depths (i.e. > 100 m, < 140 m deep), a pump profiling method was used

**BGD**

doi:10.5194/bg-2015-533

**Rates and pathways of CH₄ oxidation**

A. Sturm et al.

**BGD**

doi:10.5194/bg-2015-533

**Rates and pathways of CH$_4$ oxidation**

A. Sturm et al.

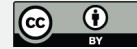

(Jones et al., 2011) in which water was pumped from depth using a double-cone intake (Jørgensen et al., 1979) sampling a thin (< 2 cm) horizontal layer of water. The intake was fastened to a conductivity, temperature, depth (CTD) probe (Sea and Sun Technology), which allowed for very accurate ($\sim$ 10 cm) vertical sample positioning during

pumping. A minimum of three tubing volumes of water was flushed through the tubing and pump system before sampling commenced. Oxygen concentrations, conductivity and temperature were determined using the CTD equipped with the following sensor array: temperature (SST PT100, accuracy $\pm$ 0.005 °C, precision $\pm$ 0.001 °C), conductivity (SST 7-pole platinum cell, accuracy $\pm$ 0.005 mS cm$^{-1}$, precision$\pm$0.0001 mS cm$^{-1}$),

$O_2$ (Oxyguard Ocean (2009), Oxyguard Profile (2010), detection limit $\sim$ 1 % saturation, precision $\pm$ 1 % saturation), light (LICOR PAR Sensor 193 SA, Accuracy $\pm$ 5 %, detection limit 0.01 µmol m$^{-2}$ s$^{-1}$), and turbidity (Seapoint). In addition to the measurements made using the Oxyguard sensors on the CTD, $O_2$ concentrations were also determined using the classic Winkler titration (detection limit 6 µmol L$^{-1}$) (APHA, 1985; Rose

and Long, 1988). The position of the oxycline was verified independently in pumped water with surface based measurements using potentiometry with Clark-style microelectrodes (Unisense) (detection limit 0.2 µmol L$^{-1}$), and by voltammetry using Au/Hg amalgam microelectrodes (1 µmol L$^{-1}$) (Brendel and Luther, 1995) (data not shown). Samples for measurement of $Fe^{2+}$ concentrations were withdrawn directly from the

pump stream or the spout of the Niskin bottle and fixed on site with ferrozine reagent, stored refrigerated (4 °C), and analyzed by a standard spectrophotometric method (Stookey, 1970; Viollier et al., 2000) within 8 h. Nitrite and $NO_3^-$ combined $NO_x$, was determined by chemiluminescnce (Braman and Hendrix, 1989) and $SO_4^{2-}$ by ion chromatography (Dionex ICS 1500, with an IONpac AS22 anion column and suppressor).

Samples for $CH_4$ concentrations were withdrawn directly from the Niskin bottle spout or pump tubing via a syringe and needle and introduced directly into pre-evacuated serum bottles with thick, butyl rubber stoppers, which were poisoned immediately with 100 µL of concentrated $HgCl_2$ (Crowe et al., 2011) to prevent methanogenesis or $CH_4$ oxidation. Methane concentrations were measured by gas chromatography (Agilent

Technologies Network GC System 6890N), and depending on concentration, using either Thermal Conductivity Detection (TCD) or Flame Ionization Detection (FID).

## 2.2 Incubations

### 2.2.1 General incubation setup

Water for incubations was sampled from the lake as described above and transferred directly into 100 mL glass syringes (110, 115, 120, and 130 m) or 60 mL plastic syringes (20, 39.5, 81.7, and 105.2 m). Here, it is important to note that the glass syringes were used for the sample depths that were evidently anoxic or in the oxic–anoxic transition zone such as the 110 m depth sample. The plastic syringes were used only for oxic depth intervals. All syringes were flushed with three volumes of water from the depth of interest and were filled without gas headspace. The openings of the syringes were closed with long (1.5″) 25 gauge needles, which were plugged with high-density butyl rubber stoppers. The $O_2$ concentration was measured inside the syringes immediately following sampling with a standard Clark-style $O_2$ electrode, inserting it into the luer-lock opening of the syringe. Oxygen was below detection (250 nmol $L^{-1}$) in the syringes from depths 110 m through to 130 m.

Radiolabeled $CH_4$ dissolved in water (2.3–4.5 $n$Ci/incubation, 5–10 µL of 10 µmol $L^{-1}$ $CH_4$ at an activity of 45.04 Ci mol$^{-1}$) was injected into the syringes by inserting a long needle into the luer-lock tip of the syringe in order to prevent the introduction of atmosphere. The syringes were closed immediately after the label addition, and a small amount of water was expelled to insure that there were no air bubbles trapped in the needle closure. The syringes were then incubated under water in the dark at an ambient temperature of ∼ 28 °C for 6 to 18 days, with sampling intervals extending from between 30 min and several hours initially to several days at the end.

Incubation sampling was conducted by removing the rubber stopper on the end of the syringe needle, flushing a small amount of water out of the needle, and then inserting the needle into an evacuated 12.5 mL Exetainer containing 1 mL of 4 M NaOH.

Discussion Paper | Discussion Paper | Discussion Paper | Discussion Paper |

**BGD**

doi:10.5194/bg-2015-533

**Rates and pathways of CH₄ oxidation**

A. Sturm et al.

**BGD**

doi:10.5194/bg-2015-533

**Rates and pathways of CH$_4$ oxidation**

A. Sturm et al.

Discussion Paper | Discussion Paper | Discussion Paper | Discussion Paper

Once the Exetainer pressure was at equilibrium with the syringe ($\sim$ 1 atm), the needle was removed, plugged with the butyl rubber stopper, and the syringe immediately returned to the water bath. At the end of the incubation period, the O$_2$ concentration was again measured inside the syringe with a Clark-style O$_2$ electrode. Oxygen was below detection (250 nmol L$^{-1}$) in the syringes sampled from 110 m through to 130 m. Exetainers were stored upside down and refrigerated at 4 °C except during air transport from Indonesia to Denmark.

### 2.2.2 Oxygen introduction to glass syringes

To estimate the possible introduction of O$_2$ into the glass syringes by diffusion along the ground glass syringe piston over the incubation period, we computed the O$_2$ fluxes along the syringe walls. Fluxes were calculated according to Fick's first law:

$$J = D\frac{\partial O_2}{\partial x} \tag{1}$$

where $J$ is the O$_2$ flux, $D$ is the calculated O$_2$ diffusivity ($2.55386 \times 10^{-5}$ cm$^2$ s$^{-1}$) (Wilke and Chang, 1955; Reid et al., 1977), $\partial[O_2]$ is the difference in O$_2$ concentration between the outside and inside of the syringe, and $\partial x$ the distance along the piston from the outside of the syringe to the water inside. The piston glass is ground along the whole length of insertion, increasing the diffusion path length $\partial x$ every time the piston was pushed further into the syringe when extracting sample (Table S1 in the Supplement). The value for $\partial[O_2]$ was set as a constant, assuming water in the syringes was anaerobic and water outside of the syringes was at saturation with respect to the atmosphere (250 µmol L$^{-1}$). The variable $\partial x$ depended on the volume remaining in the syringe and ranged from 55 to 153 mm, at 100 and 0 mL syringe volume, respectively. The total O$_2$ flux into the syringes was calculated by the product of $J$ and the area estimated between the piston wall and the inside of the syringe. Since tolerances for our syringes were not obtainable, this area was conservatively estimated by multiplying the available tolerances (0.0065 mm, Northern Technology and Testing®:

http://www.nttworldwide.com/docs/specs100ml.pdf) for glass syringes by 30 to yield an area of 21.5 mm$^2$. Summaries of possible $O_2$ introduction rates are given in Tables S1, S2 and Figs. S1, S2 in the Supplement. Introduction of $O_2$ into the glass syringes could have occurred at a maximum flux of 11.6 nmol m$^{-2}$ s$^{-1}$, with the potential to oxidize $CH_4$ at a maximum rate of 215 nmol L$^{-1}$ d$^{-1}$.

### 2.2.3 Oxygen introduction to plastic syringes

To estimate the possible $O_2$ introduction into the plastic syringes over the incubation period, we computed the $O_2$ fluxes through the plastic syringe walls. These fluxes were also calculated using Fick's first law with $O_2$ diffusivity through the polyethylene syringes of $D = 4.60 \times 10^{-8}$ cm$^{-2}$ s$^{-1}$ (Trefry and Patterson, 2001). Different initial $O_2$ concentrations produce different $O_2$ gradients for each sample depth, which translate to different diffusion rates. The value for $\partial[O_2]$ was set as a constant with the outside of the syringe at saturation with respect to the atmosphere (250 µmol L$^{-1}$), and the inside of the syringe set at the measured water column $[O_2]$ for the relevant depth intervals. The $O_2$ concentrations were expected to remain the same over the duration of the $CH_4$ oxidation rate measurements owing to low total respiration rates in Lake Matano (SA Crowe, unpublished data). Diffusion path length $\partial x$ was set to 0.1 cm, the wall thickness of the syringe. The total flux into the syringe was calculated by multiplying $F$ by the area of the syringe, which varied linearly from 99.66 to 29 cm$^2$, with a full 60 and 10 mL of water in the syringe, respectively. Summaries of $O_2$ introduction into the plastic syringes are provided in the Supplement and ranged from 0.97 µmol d$^{-1}$ (105 m, $T_o$) to 0.11 µmol d$^{-1}$ (20 m, $T_F$).

### 2.2.4 Measurement of radioactivity

Samples for the oxidation rate measurements were processed according to previously established methods (Iversen and Blackburn, 1981). Methane was flushed out of the Exetainer for 30 min at a flow-rate of 17 mL min$^{-1}$ via two needles, one needle sup-

**BGD**

doi:10.5194/bg-2015-533

**Rates and pathways of CH$_4$ oxidation**

A. Sturm et al.

Interactive Discussion

Discussion Paper | Discussion Paper | Discussion Paper | Discussion Paper

plying a $CH_4$ and air carrier gas mix (2 % $CH_4$) into the Exetainer and the other leading to a copper catalyst housed in a tube furnace, where the $CH_4$ was oxidized at 850 °C. The $CO_2$ produced was trapped directly in 20 mL scintillation vials filled with 2-phenylethylamine (4 mL) and methanol (4 mL), which has the capacity to capture 2

orders of magnitude more $CO_2$ than was produced by the $CH_4$ combustion. Immediately following $CO_2$ capture, 10 mL of scintillation cocktail (Ultima Gold XR, Packard) was added to the vials, which were vortexed for 1 min. The samples were then left to stand for 24 h before they were counted on the scintillation counter (Perkin-Elmer-Tri-Carb 2910 TR). Methane oxidation efficiency tests were performed and resulted in

a $CH_4$ to $CO_2$ conversion greater than 99.8 % at carrier-gas $CH_4$ concentrations of 2 %.

To extract the $CO_2$ from the remaining liquid in the Exetainer, 1–2 drops of bromothymol blue was added to the sample to monitor sample pH and the sample was placed into an Erlenmeyer flask along with a 20 mL scintillation vial containing a folded fiberglass filter wet with 4 mL of 2-phenylethylamine. The Erlenmeyer flask was sealed with

15 a thick, butyl rubber stopper through which a long needle was inserted to inject 2 mL of hydrochloric acid (6 M) into the Exetainer to exsolve the $CO_2$. The pH was checked visually after 24 h, and the samples were left for a total of 48 h to allow the complete exsolution of $CO_2$ and its effective entrapment on the phenylethylamine soaked filter. The scintillation vials were removed from the Erlenmeyer flasks, and 4 mL of methanol was

20 added to dissolve the precipitate, after which 10 mL of scintillation cocktail was added and the entire contents vortexed for 2 min. These samples were then left to stand for 24 h before scintillation counting. To measure the assimilation of $^{14}C$ carbon from $CH_4$ during the incubations, 3 mL of the residual fluid from each Exetainer vial was transferred into a scintillation vial and 10 mL of Scintillation cocktail added. The sample was

25 vortexed for 1 min, after which it stood for 24 h before counting. Methane oxidation and assimilation were determined from concentrations and activities using Eqs. (2) and (3). Methane oxidation and assimilation rates were then computed as first order rates from

the linear portion of the time series with two or more intervals.

$$[CH_{4oxidized}] = [CH_4]^{14}CO_2/^{14}CH_4 \tag{2}$$

$$[CH_{4assimilated}] = [CH_4]^{14}C_{residual}/^{14}CH_4 \tag{3}$$

All data used in rate calculations are tabulated in the (Table S3 in the Supplement).

## 3 Results

### 3.1 Limnology

The physical structure of Lake Matano in 2010 was much the same as previously observed (Figs. 1 and 2). During our sampling expedition, the lake was stratified with a persistent pycnocline between 110 and 250 m depth, which separates an oxic mixolimnion from a permanently anoxic monimolimnion. The mixolimnion exhibits a previously observed seasonal pycnocline (Crowe et al., 2011), which at the time of our sampling was at a depth of 30 m. A slow exchange of water across the persistent pycnocline at 110 m depth produces poorly ventilated bottom waters, which was indicated by the exhaustion of $O_2$ and the accumulation of $Fe^{2+}$. Reduced iron increases to detectable concentrations ($0.2\,\mu mol\,L^{-1}$) just below the depth at which $O_2$ becomes undetectable ($1\,\mu mol\,L^{-1}$) at 112–114 m depth (Fig. 3). Due to the rapid oxidation kinetics of $Fe^{2+}$ by $O_2$, it was unlikely that appreciable concentrations of $O_2$ coexisted with detectable $Fe^{2+}$, and we thus define the oxic–anoxic boundary as the layer between the deepest depth of detectable $O_2$ and the shallowest depth at which $Fe^{2+}$ is detected.

Most of the redox active species ($Fe^{2+}$, $Mn^{2+}$, $SO_4^{2-}$, $HS^-$) followed the classical redox cascade (Canfield and Thamdrup, 2009; Jones et al., 2011): $O_2$ and $Mn^{2+}$ overlaped slightly (Fig. 3a), which can be explained by the sluggish oxidation kinetics of $Mn^{2+}$ with $O_2$ (Jones et al., 2011; Luther, 2005; Morgan, 2005). Nitrite and $NO_3^-$ ($NO_x$) accumulated in the lower mixolimnion (Fig. 3b), with a maximum concentration

Discussion Paper | Discussion Paper | Discussion Paper | Discussion Paper |

**BGD**

doi:10.5194/bg-2015-533

**Rates and pathways of CH4 oxidation**

A. Sturm et al.

of 6 µmol L$^{-1}$ at 111 m. Fe and Mn particulates (Fig. 3c) also exhibited peaks in concentration near the pycnocline (Jones et al., 2011). Fe$^{2+}$ and NH$_4^+$ (Fig. 3a) were both below detection in the surface waters with a sharp increase at the oxic–anoxic boundary and had constant deep-water concentrations below approximately 250 m depth.

## 3.2 Oxic water column

Methane concentrations in the oxic water column were 0.5 µmol L$^{-1}$ and were oversaturated with respect to the atmosphere. Methane consumption during the 20, 39.5, and 81.7 m incubations was negligible (0.53, 0.65, and 0.13 %) for the time interval in which the rates were measured (Table 2). Rates of dissimilatory CH$_4$ oxidation (CH$_4$ oxidation), where CH$_4$ is converted into CO$_2$, between 20 and 39.5 m range from (0.2 to 0.43 nmol L$^{-1}$ d$^{-1}$) in the fully oxygenated mixolimnion (Fig. 4) and remained almost the same at 81.7 m with 0.37 nmol L$^{-1}$ d$^{-1}$. Rates of CH$_4$ assimilation into organic matter (CH$_4$ assimilation) ranged from 0.13 to 2.43 nmol L$^{-1}$ d$^{-1}$ in the mixolimnion, and summing the rates of CH$_4$ assimilation with the rates of dissimilatory CH$_4$ oxidation yielded total CH$_4$ consumption rates (total consumption) of 0.36 to 2.51 nmol L$^{-1}$ d$^{-1}$. Oxygen concentrations in the upper water column were near saturation with respect to the atmosphere and were above 200 µmol L$^{-1}$, though O$_2$ became depleted with increasing depth and concentrations were 138 µmol L$^{-1}$ at 39.5 m and 58.4 µmol L$^{-1}$ at 81.5 m (Table 2).

## 3.3 Oxic–anoxic transition layer

Rates of CH$_4$ oxidation within this layer were between 78 nmol L$^{-1}$ d$^{-1}$ at 105 m depth and 527 nmol L$^{-1}$ d$^{-1}$ at 110 m depth; rates of CH$_4$ assimilation ranged from 142 to 3640 nmol L$^{-1}$ d$^{-1}$ (105–110 m). Summing the rates of CH$_4$ assimilation and dissimilatory CH$_4$ oxidation yielded total CH$_4$ consumption in the oxic–anoxic transition layer that ranged from 220 to 4160 nmol L$^{-1}$ d$^{-1}$ (Fig. 4). Average O$_2$ concentrations

**BGD**

doi:10.5194/bg-2015-533

**Rates and pathways of CH$_4$ oxidation**

A. Sturm et al.

Discussion Paper | Discussion Paper | Discussion Paper | Discussion Paper

within this layer were less than $3\,\mu mol\,L^{-1}$, whereas $CH_4$ concentrations ranged from $6.98\,\mu mol\,L^{-1}$ at $105\,m$ to $2.70\,\mu mol\,L^{-1}$ at $110\,m$.

## 3.4 Upper anoxic zone

With $2.6\,\mu mol\,L^{-1}$ of $Fe^{2+}$ at $115\,m$, the upper monimolimnion is strictly anoxic.
Rates of $CH_4$ oxidation in this layer ranged from $5.6\,\mu mol\,L^{-1}\,d^{-1}$ at $115\,m$ depth to $50.1\,\mu mol\,L^{-1}\,d^{-1}$ at $130\,m$ depth. Rates of $CH_4$ assimilation were $6.3\,\mu mol\,L^{-1}\,d^{-1}$ at $115\,m$ and $67.2\,\mu mol\,L^{-1}\,d^{-1}$ at $130\,m$. Total $CH_4$ consumption at these depths was $11.9\,\mu mol\,L^{-1}\,d^{-1}$ at $115\,m$ and $117.4\,\mu mol\,L^{-1}\,d^{-1}$ at $130\,m$. Methane concentrations over the 115 to 130 m depth interval ranged from 12 to $484\,\mu mol\,L^{-1}$, $\cdot NO_x$ concentrations were 0.06 to $0.5\,\mu mol\,L^{-1}$, and Fe and Mn oxyhydroxide concentrations were 95–$170\,nmol\,L^{-1}$ and $1$–$361\,nmol\,L^{-1}$, respectively. Sulfate concentrations ranged from 0.6 to $19.6\,\mu mol\,L^{-1}$, and $SO_4^{2-}$ reduction rates were on the order of $0.93$–$4.73\,nmol\,L^{-1}\,d^{-1}$ over the 115 to 130 m depth interval.

## 4 Discussion

### 4.1 Oxidation in the oxic water column

Methane oxidation rates from a variety of well-oxygenated environments ($>25\,\mu mol\,L^{-1}$) have been compiled (Table 1) to provide a comparison to the rates measured in Lake Matano. The upper mixolimnion maintained $CH_4$ oxidation rates in the lower range compared to other aquatic environments (Table 1). The kinetics of aerobic microbial $CH_4$ oxidation are typically described by a Michaelis–Menten model with half-saturation constants for $CH_4$ typically between 2 and $26\,\mu mol\,L^{-1}$ (Buchholz et al., 1995; Harrits and Hanson, 1980; Lidstrom and Somers, 1984; Remsen et al., 1989) and for $O_2$ 0.14 to $21\,\mu mol\,L^{-1}$ (Joergensen, 1985; Lidstrom and Somers, 1984). With $CH_4$ concentrations below $K_m$ values, $CH_4$ oxidation rates in the mixolimnion were likely lim-

Discussion Paper | Discussion Paper | Discussion Paper | Discussion Paper | Discussion Paper

**BGD**

doi:10.5194/bg-2015-533

**Rates and pathways of CH4 oxidation**

A. Sturm et al.

**BGD**

doi:10.5194/bg-2015-533

**Rates and pathways of CH$_4$ oxidation**

A. Sturm et al.

ited by CH$_4$ availability. Though CH$_4$ concentrations in Lake Matano's mixolimnion were low, ranging from 0.5 to 3 µmol L$^{-1}$, and below typical $K_m$ values for methanotrophs, they are still well above threshold in vitro concentrations (50 to 150 nmol L$^{-1}$)(Schmidt and Conrad, 1993; Whalen et al., 1990), below which CH$_4$ oxidizing microbes can no longer access dissolved CH$_4$. High O$_2$ concentrations (> 120 µmol L$^{-1}$, or > 48 % of saturation) are also known to inhibit CH$_4$ oxidation (Harrits and Hanson, 1980) and thus at 20 m depth, CH$_4$ oxidation may have also been inhibited by as much as 58 % due to high O$_2$ concentration. However, below 45 m (< 120 µmol L$^{-1}$ O$_2$) inhibition due to O$_2$ would be unlikely (Harrits and Hanson, 1980).

## 4.2   Oxidation in the oxic–anoxic transition layer

Rates for CH$_4$ oxidation at 105 and 110 m (Fig. 4), in the oxic–anoxic transition layer, were higher than in the oxic zone. These rates were within the range reported for comparable freshwater systems (Table 1). In all systems in Table 1, O$_2$ concentrations were reported between 2.5–25 µmol L$^{-1}$, and CH$_4$ concentrations were in the low µmol L$^{-1}$ range. Lake Matano's rates of CH$_4$ oxidation in the oxic–anoxic transition layer averaged 219 nmol L$^{-1}$ d$^{-1}$. This is in the low to mid portion of the range observed in other freshwater bodies, which varies 9 orders of magnitude (Table 1). Since both O$_2$ and CH$_4$ concentrations were low within the oxic–anoxic transition zone, the rates were likely dictated by the co-availability of O$_2$ and CH$_4$. This is supported by laboratory-based studies of CH$_4$ oxidation kinetics (Buchholz et al., 1995; Harrison, 1973; Joergensen, 1985; Lidstrom and Somers, 1984).

Given the low O$_2$ concentrations in this depth interval we sought to constrain the possible availability of alternative electron acceptors for CH$_4$ oxidation through a mass balance approach. Based on mass balance considerations, the O$_2$ present at 105 m was sufficient to support the measured rates. At 110 m depth, O$_2$ was below the detection limit (0.25 µmol L$^{-1}$) of our O$_2$ sensor. Over the time interval of the rate determination, 0.114 µmol L$^{-1}$ O$_2$ would have been sufficient to oxidize the CH$_4$ con-

**BGD**

doi:10.5194/bg-2015-533

**Rates and pathways of CH$_4$ oxidation**

A. Sturm et al.

sumed, and therefore if $0.25\,\mu\text{mol}\,\text{L}^{-1}$ O$_2$ was present, yet under the detection limit of our microsensor measurements, this would provide sufficient O$_2$ (Table 2). Our calculations also suggest rates of O$_2$ diffusion into the syringe of between $1.16 \times 10^{-6}$ and $4.21 \times 10^{-7}\,\mu\text{mol}\,\text{cm}^{-2}\,\text{s}^{-1}$, could have supplied up to 19 % of the total O$_2$ needed to match the observed CH$_4$ oxidation. We argue, then, that no alternative electron acceptors were needed to support the observed rates of CH$_4$ oxidation in the oxic–anoxic transition layer. Nevertheless, total CH$_4$ consumption at the 110 m depth was $0.448\,\mu\text{mol}\,\text{L}^{-1}$, with $0.391\,\mu\text{mol}\,\text{L}^{-1}$ attributed to assimilation. If, however, assimilated CH$_4$ was also oxidized during the assimilation process, the O$_2$ available would not be sufficient to oxidize this CH$_4$, and other electron acceptors would need to be considered. Measurements for particulate Fe and Mn were not taken at this depth, but if concentrations are similar to samples below and above this depth, combined concentrations should have been between 30 and $300\,\text{nmol}\,\text{L}^{-1}$ and can therefore only account for a fraction of the oxidation needed. Sulfate and NO$_x$ were both present in quantities that would satisfy the demand for electron acceptors, but SO$_4^{2-}$ reduction rates of only a few $\text{nmol}\,\text{L}^{-1}\,\text{d}^{-1}$ rule out its involvement in CH$_4$ oxidation of this magnitude, leaving NO$_x$ as the likely electron acceptor at the depth of 110 m.

## 4.3 Oxidation in the anoxic waters

Methane oxidation was also observed in incubations of water from depths (115, 120, and 130 m) below the oxic–anoxic transition zone (Fig. 4). This observation requires us to carefully consider the possible oxidants available. As above, we first consider possible O$_2$ contamination by diffusion into the syringes, which ranged from $1.16 \times 10^{-6}$ to $7.28 \times 10^{-7}\,\mu\text{mol}\,\text{O}_2\,\text{cm}^{-2}\,\text{s}^{-1}$ (Table S1 in the Supplement) and can contribute at most 1.86, 11.8, and 0.216 % at 115, 120, and 130 m, respectively to the observed CH$_4$ oxidization (Table 2). There are several other theoretically possible oxidants (SO$_4^{2-}$, NO$_2^-$, NO$_3^-$, Fe$^{3+}$, Mn$^{4+}$) for which reactions are given in Table 3.

Discussion Paper | Discussion Paper | Discussion Paper | Discussion Paper |

**BGD**

doi:10.5194/bg-2015-533

**Rates and pathways of CH$_4$ oxidation**

A. Sturm et al.

Interactive Discussion

To validate if these electron acceptors are suitable, we have calculated $\Delta_r G$ values for each electron acceptor considering the in situ conditions in Lake Matano. The possible Gibbs free energy space defined by substrate concentrations in Lake Matano is marked in green on the graphs in Fig. 5 along with energies calculated specifically for the three anoxic samples at 115, 120, and 130 m depth. The conditions at which the $\Delta G$'s are $-30$ and $-15$ kJ mol$^{-1}$ (Iso-energies) are marked in blue and red, identifying the lowest energies necessary for the generation of ATP and energies observed in similar environments known to support microbial growth, respectively (Caldwell et al., 2008; Hoehler et al., 1994; Valentine and Reeburgh, 2000). These boundaries, hence, are the lowest known amounts of energy at which organisms can grow using these metabolic pathways. All electron acceptors considered provide more than this minimum amount of free energy, except for $SO_4^{2-}$, which is close to the minimum energies (Fig. 5).

Concentrations of electron acceptors and their possible contribution to CH$_4$ oxidation in %, based on the reactions listed in Table 3, are given in Table 2 for each depth. Nitrite and $NO_3^-$ are combined as $NO_x$, and $Fe^{3+}$ and $Mn^{4+}$ are referred to as $Mn_{part}$ and $Fe_{part}$, since they primarily exist as particles in these oxidation states. The concentrations of available reactive $Fe^{3+}$ and $Mn^{3+}/Mn^{4+}$ were low, approximately 250–530 nmol s L$^{-1}$ cumulative between 115 and 120 m and less than 100 nmol s L$^{-1}$ at 130 m, compared to the amount of CH$_4$ oxidation observed, and could only account for up to 7.7, 22.1, and 0.37 %, respectively, of the CH$_4$ oxidation measured (Table 2). Nitrate-nitrite contributions to the oxidation of CH$_4$ would be low at 115 m and considerable at 120 m with 7.6 and 64 %, and then sink to a negligible 0.8 % at the 130 m depth. The only other significant oxidant at 115 and 120 m was $SO_4^{2-}$, which could account for all the CH$_4$ oxidation observed at these depths, though at 130 m, $SO_4^{2-}$ was below our detection ($> 250$ nmol L$^{-1}$), a concentration at which $SO_4^{2-}$ could only contribute 7.5 % to the CH$_4$ oxidation observed (Table 2). Similar to the oxic–anoxic transition layer, despite the apparent availability of $SO_4^{2-}$ as an electron acceptor, the measured $SO_4^{2-}$ reduction rates at these depths were less than 5 nmol L$^{-1}$ d$^{-1}$ (Fig. 4c). This suggests

that $CH_4$ oxidation coupled to $SO_4^{2-}$ reduction alone cannot support the observed rates of AOM between the depths of 115 to 130 m. We are thus left with some uncertainty as to specific electron acceptors involved in $CH_4$ oxidation at Lake Matano given our inability to constrain the electron mass balance. It is likely, however, that we have under-
estimated the concentrations of particulate Fe and Mn (Jones et al., 2011). Particles too small to be retained on 0.2 μm filters, for example, may have escaped measurement. Such small particles have been observed in the lake as nano-particulate aggregates containing Fe and Mn using SEM and TEM (Jones et al., 2011; Zegeye et al., 2012) and would be among the most reactive of all the particles in the system. Though $O_2$ concen-
trations were measured in the syringes and all were below our DL (250 nmol $L^{-1}$), $O_2$ contamination could have occurred during sample handling but evaded detection due to its consumption by $Fe^{2+}$ prior to measurement. For example, if sampling introduced 1–2 μmol $L^{-1}$ $O_2$ this could have been rapidly consumed through oxidation of $Fe^{2+}$, sup­plying 4 to 8 μmol $L^{-1}$ additional reactive Fe oxyhydroxides. Regardless of the uncer-
tainties in available electron acceptors, the potential rates of anaerobic $CH_4$ oxidation measured are high, even in comparison to rates measured in other fresh (Iversen et al., 1987; Joye et al., 1999) and marine water columns (Wankel et al., 2010).

## 4.4  $CH_4$ assimilation

In Lake Matano a large fraction of $CH_4$ metabolized (up to 87 %) was assimilated
(Fig. 6). The fraction of $CH_4$ carbon assimilated in Lake Matano is indeed as high as values reported in early isolation and characterization laboratory studies (80 % and 47–70 %) (Brown et al., 1964; Vary and Johnson, 1967). Most studies of temperate lakes report that about one-third of the carbon from $CH_4$ consumption was assimilated with two-thirds oxidized to $CO_2$ (Hanson, 1980; Harrits and Hanson, 1980; Lidstrom and
Somers, 1984; Rudd et al., 1974). For example, the temperate meromictic Lake 120 of the Experimental Lake Area of Northern Ontario (Rudd et al., 1974), which, like Lake Matano, is persistently stratified, exhibited only 36.8 % assimilation. Perhaps because

Discussion Paper | Discussion Paper | Discussion Paper | Discussion Paper |

**BGD**

doi:10.5194/bg-2015-533

**Rates and pathways of CH₄ oxidation**

A. Sturm et al.

Discussion Paper | Discussion Paper | Discussion Paper | Discussion Paper | Discussion Paper

of its physical and chemical similarities, the most appropriate comparison is Lake Tanganyika, which exhibits on average 26 % assimilation (Rudd, 1980). We suggest that the high percentage of assimilation in Lake Matano may reflect its highly oligotrophic nature. In more productive systems, excess organic carbon is available for growth, thus

most $CH_4$ carbon is channeled through dissimilation to $CO_2$ (Rudd, 1980). Anecdotally, carbon assimilation from $CH_4$ in marine systems is generally higher than in freshwater systems (Ward et al., 1987) and marine systems are typically more oligotrophic than freshwater.

These relative high rates of $CH_4$ assimilation suggest that $CH_4$ may be an important

source of carbon for biomass in Lake Matano. Though rates of $CH_4$ assimilation within the surface waters (10 m, $0.2\,\mathrm{nmol\,L^{-1}\,d^{-1}}$) are small compared to dark carbon fixation rates ($14\,\mathrm{nmol\,L^{-1}\,d^{-1}}$), the deeper anoxic waters exhibit $CH_4$ assimilation rates of up to $8.2\,\mathrm{\mu mol\,L^{-1}\,d^{-1}}$ (120 m) which exceeds dark carbon fixation rates of $0.24\,\mathrm{\mu mol\,L^{-1}\,d^{-1}}$ (at 117.5 m) by 34 times, implicating $CH_4$ assimilation as the largest carbon fixing pro-

cess within this depth interval.

## 5   Conclusions

Despite the accumulation of $CH_4$ to relatively high concentrations in Lake Matano's deep monimolimnion, much of this $CH_4$ is oxidized directly within the lake, precluding strong $CH_4$ fluxes to the atmosphere. Lake Matano, indeed, supports relatively high

rates of $CH_4$ oxidation with some $CH_4$ apparently oxidized anaerobically (for example, rates of oxidation up to $5.6\,\mathrm{\mu mol\,CH_4\,L^{-1}\,d^{-1}}$ occur at 115 m depth in the apparent absence of $O_2$). Alternative electron acceptors that may support measured AOM include $NO_x$, the oxidized forms of Fe and Mn, and $SO_4^{2-}$. Measured rates of $SO_4^{2-}$ reduction are, however, too low to support the measured rate of $CH_4$ oxidation, and we favor

$NO_x$ and oxidized Fe and Mn as the likely electron acceptors. Lake Matano also exhibits unusually high rates of $CH_4$ assimilation, which may be related to its oligotrophic

# BGD

doi:10.5194/bg-2015-533

**Rates and pathways of CH$_4$ oxidation**

A. Sturm et al.

nature. Regardless, such high rates of assimilation and oxidation implicate $CH_4$ as an important carbon and energy source for microbial growth in Lake Matano. By extension to other oligotrophic ferruginous environments, $CH_4$ was likely an important contributor to microbial metabolisms in Earth's ferruginous Precambrian oceans.

**The Supplement related to this article is available online at doi:10.5194/bgd-13-1-2016-supplement.**

*Author contributions.* A. Sturm and S. A. Crowe designed the Experimental setup and executed the experiment; they were also responsible for the data acquisition and analysis, interpretation and manuscript preparation.
D. A. Fowle was involved in inexperimental design, scientific discussion, data interpretation and manuscript preparation.
C. A. Jones conducted fieldwork and contributed to manuscript preparation.
K. Leslie helped measuring methane concentrations and reducing and interpretation methane concentration data, she also contributed to manuscript preparation.
D. E. Canfield was involved in scientific discussions, data interpretation and manuscript preparation. S. Nomosatryo and C. Henny provided logistical and field sampling support in Indonesia.

*Acknowledgements.* The authors thank Alfonso Mucci and Bjørn Sundby, for inspiring discussions, D. Rahim, and S. Rio are acknowledged for sampling and logistical support in Indonesia. B. Thamdrup for lending a tube furnace for $CH_4$ combustions. The authors are grateful to PT-INCO Tbk for in kind support. The Danish National Research Foundation provided funding for C. Jones, S. A. Crowe, and D. E. Canfield. A. Sturm and D. A. Fowle were supported by the National Science Foundation grant EAR-0 844 250.

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

**Table 1.** Methane oxidation rates complied from a variety of global aquatic settings. Values in brackets have uncertainties with respect to oxygen concentrations during measurements.

| Lake/Reservoir | Oxidation rate µmol L$^{-1}$ d$^{-1}$ Oxic/Oxic–Anaerobic transition/Anaerobic | Source |
|---|---|---|
| Lake Mendota, (Madison, Wisconsin) | 28.8/4.8/5.8 | Harrits and Hanson (1980) |
| Lake Kivu, (Africa) | 0.48/0.43/0.89 | Jannasch (1975) |
| ELA, (Nothern Ontario) | 72/890 | Rudd and Hamilton (1975) |
| Lake Pavin, (France) | 0.006–0.046/–/0.4 | Lopes et al. (2011) |
| Lake Kasumigaura, (Japan) | –/0.12/– | Utsumi et al. (1998a) |
| Lake Nojiri, (Japan) | –/17/– | Utsumi et al. (1998b) |
| Lake Erie, (USA, Canada) | –/3.84/– | Howard et al. (1971) |
| Big Soda Lake, (Nevada) | 0.0013/0.01/0.064 | Iversen et al. (1987) |
| Mono Lake, (California) | 0.04–3.8/0.5–37/48–85 nM d-1 | Joye et al. (1999) |
| ELARP pond, FLUDEX reservoirs (ELA) | (360–1200) | Venkiteswaran and Schiff (2005) |
| Petit-Saut Reservoir, (Brazil) | 1600/–/– | Guerin and Abril (2007) |
| Lake 120 and 227, (ELA, Nothern Ontario) | 1.3/–/0.49 | Rudd et al. (1974) |
| Lake Tanganyika, (Arfrica) | 0.1–0.96/0.17–1.8/0.24–1.8 | Rudd (1980) |
| Lillsjoen lake, (Sweden) | 0.33/0.01/– | Bastviken et al. (2002) |
| Marn lake, (Sweden) | 0.81/2.17/2.2 | Bastviken et al. (2002) |
| Illersjoen lake, (Sweden) | –/–/1.3–3 | Bastviken et al. (2002) |
| Lake Kevätön, (Finland) | 27/–/– | Liikanen et al. (2002) |
| Lake Matano, (Sulawesi, Indonesia) | $3.6 \times 10^{-4}$–$2.5 \times 10^{-3}$/0.22/4.2–117 | This Study |
| Brine pool (Gulf of Mexico) | –/–/3.5 | Wankel et al. (2010) |
| Cariaco Basin, (Pacific) | $1 \times 10^{-7}$/$1 \times 10^{-6}$/$4 \times 10^{-4}$ | Ward et al. (1987) |
| Black Sea | –/$1 \times 10^{-6}$/$1.64 \times 10^{-3}$ | Kessler et al. (2006) |
| Lake Spirit, (Oregon) | 0.144/0.065/– | Lilley et al. (1988) |
| Hudson River, (New York) | $4 \times 10^{-6}$–$6 \times 10^{-4}$/–/– | Deangelis and Scranton (1993) |

**BGD**

doi:10.5194/bg-2015-533

**Rates and pathways of CH$_4$ oxidation**

A. Sturm et al.

**Table 2.** Availability of redox species in the incubation syringe in µmol L$^{-1}$, "oxidation potential (%)" is the potential contribution of each redox species towards oxidizing CH$_4$ to CO$_2$. Maximum O$_2$ introduction into syringes by diffusion and its possible increase of O$_2$ content compared to background in [%] and how much this amount of O$_2$ could have been part in the observed CH$_4$ oxidation (%). Methane oxidation to CO$_2$ during the incubation is displayed in nmol L$^{-1}$ and as % of total CH$_4$ available, along with total CH$_4$ consumed.

| Incubation Depth (m) | Conc. in µmol L$^{-1}$ (%) contribution to CH$_4$ oxidation | | | | | Max O$_2$ diffusion into Syringes in µmol L$^{-1}$, O$_2$ increase wrt background conc. [%] and its CH$_4$ oxidation Potential (%) | CH$_4$ converted to CO$_2$/CH$_4$ total nmol L$^{-1}$ (%) |
|---|---|---|---|---|---|---|---|
| | Fe$_{part}$ | Mn$_{part}$ | SO$_4^{-2}$ | NO$_3^-$ and NO$_2^-$ | O$_2$ | | |
| 20 | 0.077 (452) | 0.003 (35.2) | 26.52 (1.25 $\times10^6$) | 0.4 (9.4 $\times10^3$) | 210.5 (4.94 $\times10^6$) | 45 [22] (1.06 $\times10^6$) | 2.13/3.84; (0.30/0.53) |
| 39.5 | no Data | 0.003 (15.8) | 28.28 (5.95 $\times10^5$) | 0.85 (9.0 $\times10^3$) | 137.2 (1.44 $\times10^6$) | 129 [94] (1.35 $\times10^6$) | 4.59/5.98; (0.50/0.65) |
| 81.7 | no Data | 0.003 (197.4) | 24.75 (6.5 $\times10^6$) | 4.32 (5.7 $\times10^5$) | 58.4 (7.68 $\times10^6$) | 13 [23] (1.79 $\times10^6$) | 0.38/2.56; (0.02/0.13) |
| 105 | no Data | no Data | 20.44 (8.7 $\times10^4$) | 4.88 (1.0 $\times10^4$) | 4.4 (9.4 $\times10^3$) | 5 [114] (1.07 $\times10^4$) | 23.5/66.2; (0.36/1.01) |
| 110 | no Data | no Data | 18.93 (3.3 $\times10^4$) | 5.39 (4.8 $\times10^3$) | 0.25 (2.2E2) | 0.022 [2.2] (19.4) | 56.7/448; (2.1/16.6) |
| 115 | 0.107 (2.03) | 0.148 (5.62) | 31.55 (4.8 $\times10^3$) | 0.1 (7.6) | 0 (0.00) | 0.024 [–] (1.86) | 648/1408; (7.1/15.2) |
| 120 | 0.101 (20.9) | 0.003 (1.24) | 9.91 (1.6 $\times10^4$) | 0.077 (64) | 0 (0.00) | 0.014 [–] (11.8) | 60.5/732; (0.05/0.61) |
| 130 | 0.097 (0.36) | 0.001 (0.008) | 0.25 (7.5) | 0.055 (0.83) | 0 (0.00) | 0.014 [–] (0.216) | 3330/7092; (0.63/1.47) |

Discussion Paper | Discussion Paper | Discussion Paper | Discussion Paper |

**BGD**

doi:10.5194/bg-2015-533

**Rates and pathways of CH$_4$ oxidation**

A. Sturm et al.

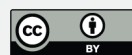

**Table 3.** Mass action equations and their corresponding Gibbs Free energy (ΔG°) of reaction at 298.15 K for demonstrated and theoretical AOM pathways.

| Reaction | $\Delta G^{0'}$ in kJ mol$^{-1}$ CH$_4$ | Reference |
|---|---|---|
| $5CH_4 + 8NO_3^- + 8H^+ \rightarrow 5CO_2 + 4N_2 + 14H_2O$ | −765 | Raghoebarsing et al. (2006) |
| $3CH_4 + 8NO_2^- + 8H^+ \rightarrow 3CO_2 + 4N_2 + 10H_2O$ | −928 | Raghoebarsing et al. (2006) |
| $CH_4 + 8Fe(OH)_3 + 15H^+ \rightarrow HCO_3^- + 8Fe^{2+} + 21H_2O$ | −572.2 | Crowe et al. (2011) |
| $CH_4 + 4MnO_2 + 7H^+ \rightarrow HCO_3^- + 4Mn^{2+} + 5H_2O$ | −789.9 | Crowe et al. (2011) |
| $CH_4 + SO_4^- \rightarrow HCO_3^- + HS^- + H_2O$ | −20 to −30 | Boetius et al. (2000); Valentine et al. (2000) |

Discussion Paper | Discussion Paper | Discussion Paper | Discussion Paper

# BGD

doi:10.5194/bg-2015-533

**Rates and pathways of CH$_4$ oxidation**

A. Sturm et al.

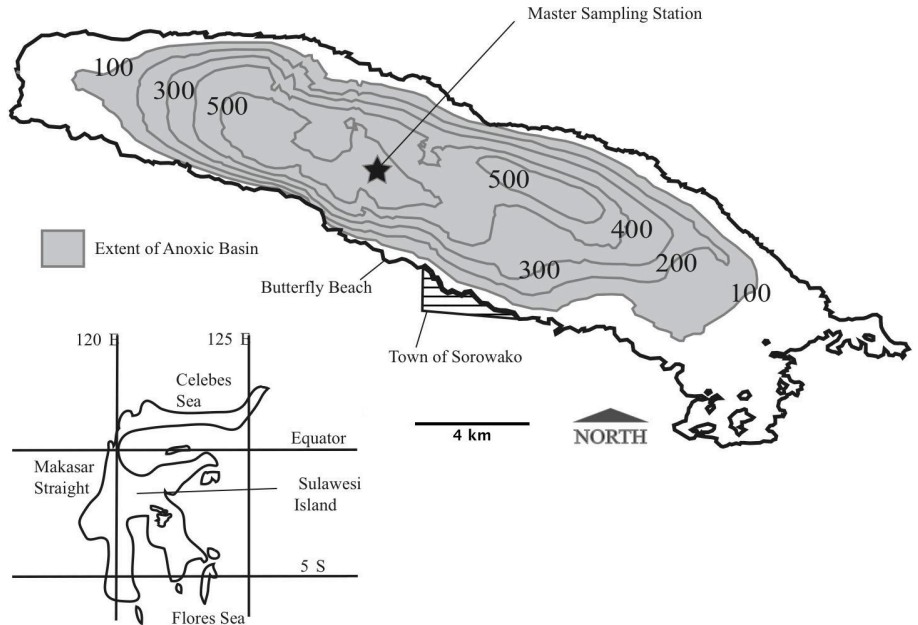

**Figure 1.** Map of Lake Matano bathymetry, with the extent of the anoxic basin is shaded. The location of the deep water master sampling station and the town of Sorowako are indicated (modified after Crowe et al., 2008).

Discussion Paper | Discussion Paper | Discussion Paper | Discussion Paper |

**BGD**

doi:10.5194/bg-2015-533

**Rates and pathways of CH$_4$ oxidation**

A. Sturm et al.

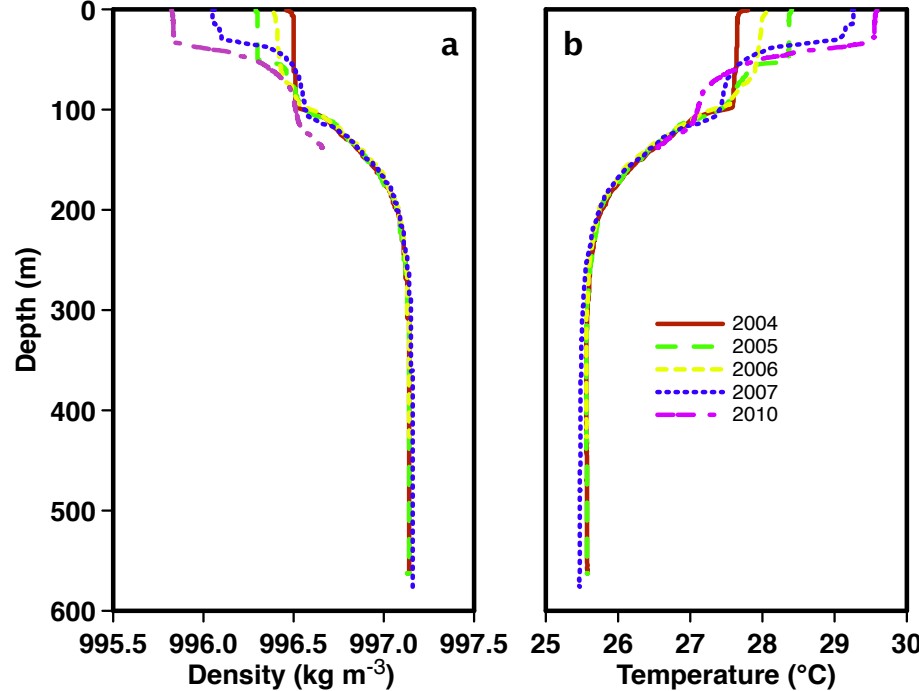

**Figure 2.** Multi-year density **(a)** and temperature **(b)** profiles for Lake Matano.

Discussion Paper | Discussion Paper | Discussion Paper | Discussion Paper

**BGD**

doi:10.5194/bg-2015-533

**Rates and pathways of CH$_4$ oxidation**

A. Sturm et al.

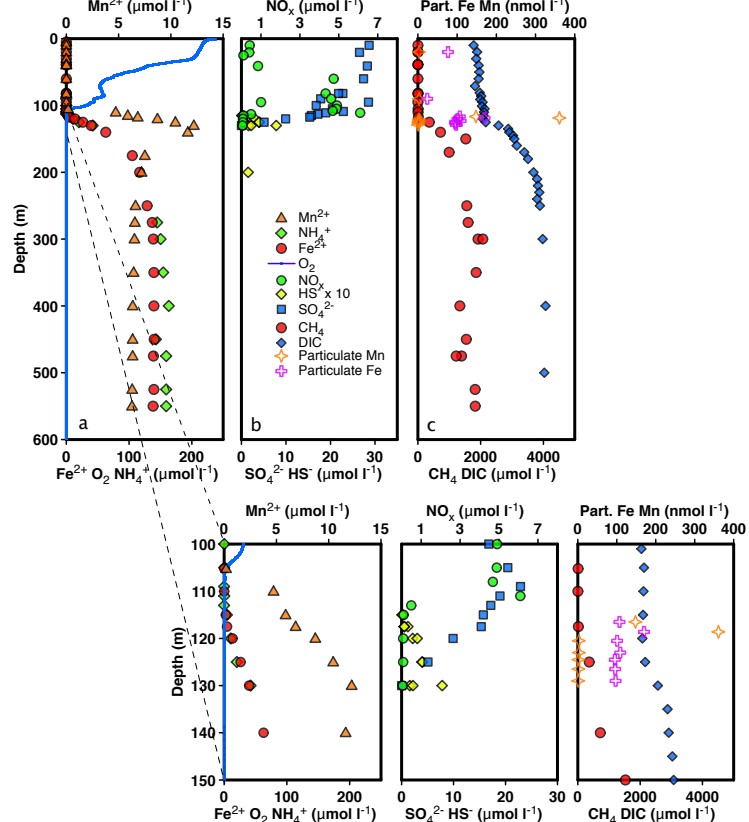

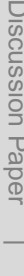

**Figure 3.** Lake Matano profiles for **(a)** O$_2$ (CTD), aqueous Fe$^{2+}$, Mn$^{+2}$ and NH$_4^+$ **(b)** aqueous NO$_x$, HS$^-$ and SO$_4^{2-}$, and **(c)** dissolved gases CH$_4$, DIC, particulate Fe and Mn (Jones et al., 2011).

Discussion Paper | Discussion Paper | Discussion Paper | Discussion Paper | Discussion Paper

**BGD**

doi:10.5194/bg-2015-533

**Rates and pathways of CH$_4$ oxidation**

A. Sturm et al.

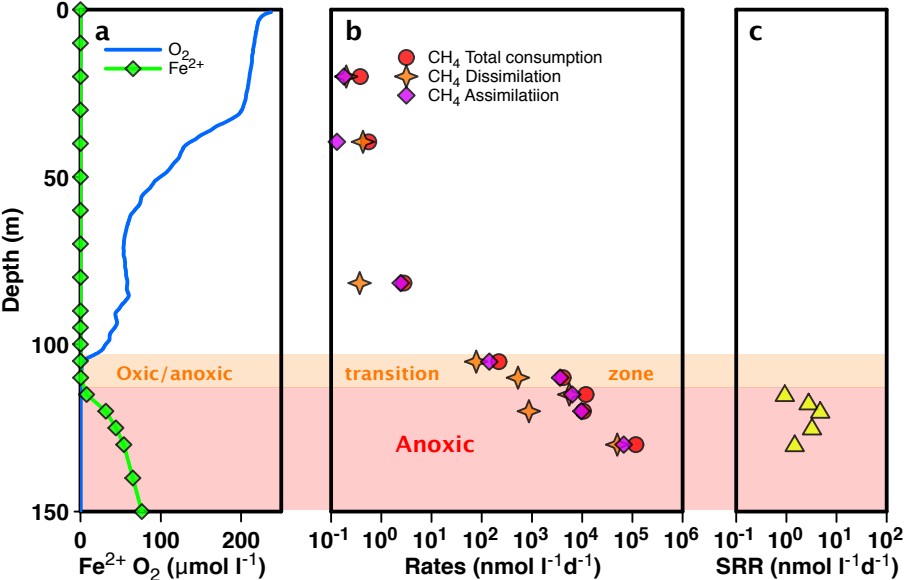

**Figure 4.** Dissolved biogeochemically active elements and process measurements as a function of depth **(a)** dissolved Fe$^{2+}$ and O$_2$; **(b)** rates of production of CO$_2$ (dissimilatory CH$_4$ oxidation), CH$_4$ assimilation and total CH$_4$ consumption during the experiment; **(c)** sulfate reduction rates.

Discussion Paper | Discussion Paper | Discussion Paper | Discussion Paper |

**BGD**

doi:10.5194/bg-2015-533

**Rates and pathways of CH$_4$ oxidation**

A. Sturm et al.

Interactive Discussion

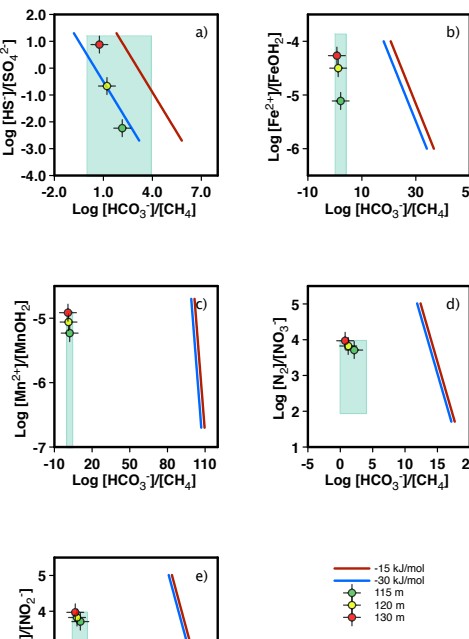

**Figure 5.** The plots show the energy availability at different concentrations of substrates and products for each reaction. The green areas show the range of substrate and product concentrations of Lake Matano, added are the specific anaerobic depth (115, 120, 130 m) at which our experiments were conducted. The "Iso-energy" lines show that all proposed pathways of methane oxidation are favorable except for **(a)** ($SO_4^{2-}$), where the $\Delta_r G$ values are very close to the previously proposed minimum values of $-15\,\mathrm{kJ\,mol^{-1}}$ for cell survival.

Discussion Paper | Discussion Paper | Discussion Paper | Discussion Paper

**BGD**

doi:10.5194/bg-2015-533

**Rates and pathways of CH$_4$ oxidation**

A. Sturm et al.

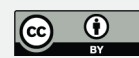

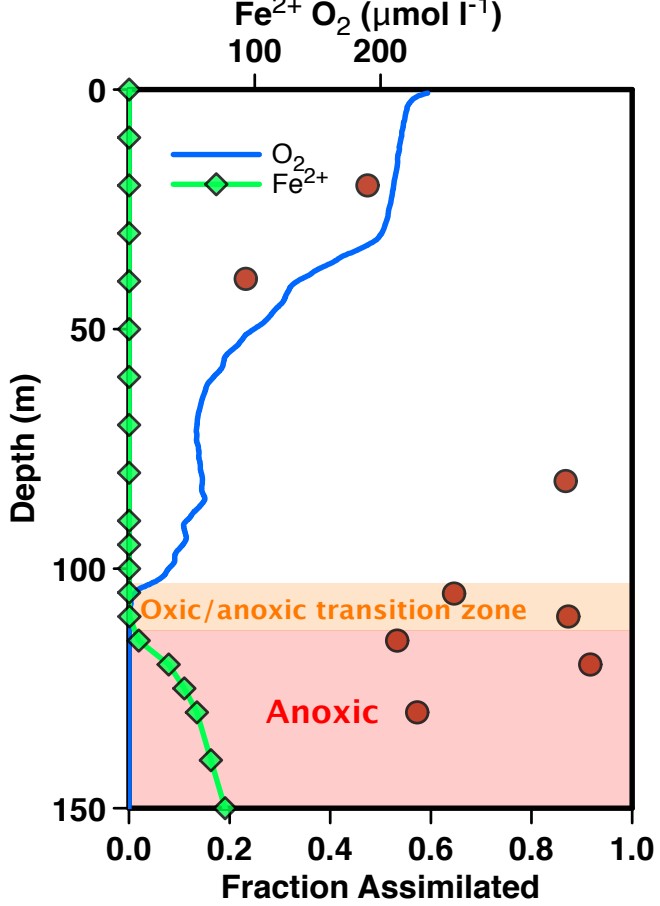

**Figure 6.** Fraction of CH$_4$ consumed through assimilation during the incubations (based on first order rate kinetics).