# Peer review of "Rates and pathways of CH4 oxidation in ferruginous Lake Matano, Indonesia"

_Biogeosciences, 2015_

## Referee Comment (RC1) · Anonymous Referee #1 · 4 Apr 2016

This review is based on files uploaded by Arne Stum (17 oct 2015)

General comment

The main subject of this manuscript is methane oxidation in the ferruginous lake Matano. The topic is really interesting, since anaerobic methane oxidation in fresh-waters is understudied. The environment is well chosen to study anaerobic methane oxidation coupled with other electron acceptors than sulfate. Just put in evidence such high rates of anaerobic CH4 oxidation is already very interesting. Also, results of the importance of CH4 in the foodweb are really interesting.

However, numerous informations are lacking. Results of anerobic methane oxidation, which is the main subject of this manuscript, are poor. Authors only measured sulfate reduction during their incubations, whereas the decrease of nitrate, nitrite, iron and

manganese concentrations in the incubations would have been really informative. Also, pyrosequencing data are missing. For example, these data would have been really interesting to understand why anaerobic methane oxidation is not coupled with sulfate reduction at 115 and 120m, whereas sulfate concentrations are clearly sufficient. Also, authors talk about CH4 fluxes in their abstract and conclusion, but no data of fluxes are shown.

General question: Oxidation in the oxic water column: Authors justify low aerobic CH4 oxidation rates by the inhibition by very high O2 concentrations above 45 m. However, below 45 m, they say that inhibition due to O2 would be unlikely. So, how to explain low aerobic CH4 oxidation rates from 45 to 100 m ? Also, how do the authors explain low CH4 concentrations in the oxic compartment, since aerobic CH4 oxidation is low, compared with huge amounts of CH4 in anoxic water column ? By the water column structure of the lake (not fully described in this manuscript), by the importance of anaerobic CH4 oxidation ?

Specific comments

Abstract Rates are lacking throughout all the abstract. Please correct. Line 4: Authors say here that methane oxidation rates are low, while they show in table 1 and in text that anaerobic methane oxidation rates are high. Please clarify. Line 5: "to preclude strong CH4 fluxes to the atmosphere" => What proof do you have ? No measurements of CH4 fluxes were made in this paper. Line 8-9: No direct evidence of which electron acceptors are used. Line 11: Please note the different fractions (fraction of CH4 assimilated and fraction of CH4 oxidized to CO2). Line 12-13: "...and potentially other ferruginous or low productivity environments" => authors did not study another environment, it is clearly speculative => not in the abstract.

Introduction

References are not well sorted. For example, Page 2 Line 17: Kroeger 2011, Cicerone 1988, Conrad 2009; not sorted by year, nor by name. Page 2 Line 17: references are

not well chosen. Please reference more general papers/reports, such as IPCC. Page 2 Line 22: not only in sediments and soils; methanogenesis can also occur in anoxic water columns Page 3 Lines 8-9: "aerobic CH4 oxidation is believed to dominate CH4 consumption in these environments" => reference ? Page 3 Line 12-13: "In the large intervals....where CH4 occurs": sentence not clear => please clarify Page 3 Line 12-14: Reference ? Page 3 Line 16-17: Reference ? Please better describe water column structure of Lake Matano => Is the structure dependent of season ? Sulfate, nitrate, nitrite... concentrations are the same throughout the year or do they change according to the season ? Etc.

Material and methods

Page 5 Line 14: (ie. <100 m or > 140m, respectively). Page 5 Lines 13-19: Why did authors use different sampling methods for deep and shallow and intermediate depths ? Page 7 Line 22: "to prevent the introduction of atmosphere" => to prevent the introduction of atmospheric O2. Page 8 Lines 1-3: "The syringes....at the end" => sentence not clear. Please clarify the incubations' times. Page 8 Lines 10-11: "Oxygen was below detection...to 130 m" => it is a result; not in the M&M. Page 9 Lines 10-13: "Summaries....215 nmol l-1 d-1." => also a result; not in the M&M. Page 10 Lines 6-8: "Summaries.... 0.11". => not in the M&M.

Results

Page 12 Lines 3-4: "as previously observed" => put the reference of Crowe's papers here. Page 12 Line 22: it's easier when the text is in the same order as figures => talk about Fe2+ and NH4+ (Fig. 3a) at the same time as O2 and Mn2+. Page 13 Line 9: between 20 m and 39.5 m ranged from 0.2 to 0.43. Page 14 Lines 7-15: references of the figures are missing.

Discussion

Page 14 Line 20: Table 1 is after table 2 in the text (table 2 appears page 13). Page 15

Line 18: "comparable freshwater systems" => all the lakes in Table 1 are not comparable to Matano. Temperature highly influences bacterial processes and numerous lakes referenced in this table are boreal or temperate lakes, while Matano is a tropical lake. Page 17 Lines 17-21: sentence not clear => please clarify. Page 18 Lines 9-10: "..., and could ONLY account for up to 7.7, 22.1 and 0.37 %..." => I find that contributing for 1/5 (22.1%) of CH4 oxidation is not negligible, so the word "only" is misused. Page 18 Lines 12-16: sentence not clear, too long => please clarify. Page 18 Lines 18-20: "This suggests... to 130m" => At 115-120 m, sulfate concentrations are clearly sufficient to explain all the CH4 oxidation observed, but sulfate reduction rates are very low. Why ? How do you explain that ? Is it due to bacterial communities ? Page 19: Authors admit that a potential contamination by O2 into syringes lead to uncertainties in the availability of the electron acceptors. I wonder if this method with syringes was well chosen. Why not use glass serum bottles ?

Figures and tables

Table 1: Mono Lake nM d-1 => rates are in nmol L-1 d-1 ? Column title: oxidation rate $\mu$mol L-1 d-1 => please clarify. Figure 2: This manuscript shows the results of the field campaign 2010. In this figure, 4 other dates are shown, with no reference to another paper in the caption. Please clarify. Figure 3: Please put the legend outside the graph b. Figure 4: The scales of rates are not useful at all. We cannot visualize the rates precisely. Also, please more divide Y scale. Figure 5: caption not clear and no complete. Figure 6: legend not complete => what are dots ? And why show Fe2+ concentrations in this figure ?

---

## Referee Comment (RC2) · Anonymous Referee #2 · 13 Apr 2016

This is an interesting paper that contributes nicely to a growing body of work suggesting an important role for anaerobic methane oxidation in terrestrial systems. I think the conclusions should be more circumspect- the data to not add up to convincing proof or disproof for the role of any major electron acceptor.

p 2 line 22 "A significant fraction of this CH4 is then consumed through microbially catalyzed oxidation". This is a scientifically improper use of "significant". Consumption of 1% could be statistically significant, but would not be meaningful.

p 3 line 9 "Large intervals of diffusion limited marine sediments contain abundant sulfate and are therefore favorable to AOM". Interval implies a space or regular spaces between two delimiters. Perhaps "zone" would be better?

p 3 line 26 "However, natural systems rich in nitrate with rapid N-cycling, such as freshwater lakes, estuaries, and wetlands, should have the potential to oxidize immense amounts of CH4 through nitrate dependent AOM (Joye et al., 1999)." The adjectives "immense" and "rich" here are probably inappropriate. Nitrogen is limiting in most terrestrial environments, and the amounts of nitrate available will generally be stoichiometrically too low to account for methane oxidation. The implication that nitrate in freshwaters is an analog of sulfate in marine waters is unreasonable. Notably, the process has only been demonstrated so far in highly polluted waters and wastewater.

On the other hand, there are a growing number of studies demonstrating substantial AOM in freshwater wetlands, although (as in the present study) most lack clear evidence of what the main electron acceptors are. The authors list a few of these studies on p 4 line 1-5, but could give a broader perspective on recent environmental studies of AOM and the estimated potential magnitude of AOM in freshwater systems. See, e.g. Gupta et al Environ. Sci. Technol., 2013, 47 (15), pp 8273–8279; Segarra et al 2015 Nature Comm 6: 7477; and several others. The referencing in general in the manuscript is biased towards older studies. Only about 20 of the 75 references were written in the last decade. The Introduction and Discussion, and Table 1, could use some updated context.

Section 2.2.2 is a bit unclear: D is the O2 diffusivity in water ($2.55386 \times 10-5$ cm2 s−1) (at what T?), but the D along a tortuous path between the piston and the syringe wall would presumably be much less than this. Why did the authors not simply perform an experiment with sterilized anoxic water to control for this leakage? The conclusion about syringe leakage in Section 4.2 p 14 lines 2-5 is worrying: "Our calculations also suggest rates of O2 diffusion into the syringe of between $1.16 \times 10-6$ and $4.21 \times 10-7$ $\mu$mol cm−2 s−1, could have supplied up to 19 % of the total O2 needed to match the observed CH4 oxidation." This is a large amount, and as we should assume that the calculation is only approximate, this is a very large potential source of error and uncertainty. Fortunately, the interesting part of the study is not the oxic water, but the anoxic waters discussed in section 4.3.

The units used to summarize the findings of sections 2.2.2 and 2.2.3 are different. It would be clearer if these were in the same units, e.g. if both were summarized in a similar way such as: "with the potential to oxidize CH4 at a maximum rate of xx nmol L−1 d−1."

Figure 5 and p 15 line 12: "All electron acceptors considered provide more than this minimum amount of free energy, except for sulfate, which is close to the minimum energies (Fig. 5)." Firstly. I disagree with the interpretation given in the figure legend that sulfate is close to the −15 kJ mol−1 threshold. It is closer to -30, and a viable process. Secondly, this actually suggests to me that sulfate is the most likely electron acceptor for methane oxidation. If methane oxidation coupled to sulfate reduction is active and methane is in excess, then the bacterial community should grow to the point where it reduces sulfate close to an equilibrium point where a minimum energy is obtained. The excess available amounts of other electron acceptors suggest that these are not effectively being used to oxidise methane.

Another issue is that the rates are not measured in situ, they are estimated in closed incubation vessels (for a period of up to 18d) after sampling the waters. Therefore the rates do not account for diffusive fluxes of the electron acceptors in situ. In the absence of diffusive flux, sulfate, which is normally near an equilibrium level, could become depleted in the incubation syringes below the level where it can support methane oxidation. The methane oxidation rates and the sulfate reduction rates may therefore both be underestimated.

I also do not fully agree with the conclusion in the Abstract line 7: "Here, CH4 oxidation proceeds in the apparent absence of oxygen (O2) and instead appears to be coupled to nitrate (NO−3 ), nitrite (NO−2), iron (Fe), or manganese (Mn) reduction." This is repeated in the conclusion. It seems to be a selective interpretation of the data. I do not believe that sulfate can be discounted, not that there is clear evidence for nitrate, Fe or Mn. The data do not seem to add up to a coherent theory supporting any single oxidant. For example, the results presented on p 15 lines 17-22 ff contradict the authors' conclusion. The calculations can account for only small amounts of measured methane oxidation via Fe, Mn or nitrate reduction, with the sole exception of nitrate at a single depth (130 m). On the other hand, sulfate concentrations could account for all the CH4 oxidation observed at these depths (except at 130 m). The argument against sulfate reduction is the low measured sufate reduction rates. However, the contradictory nature of the two lines of evidence, coupled with the absence of reported reduction rates for nitrate, Fe and Mn, does not add up to clear evidence for any single oxidant.

I think a better approach is to stress that the process of anaerobic methane oxidation occurs but to embrace the uncertainty (as on p 16 line 2) about the mechanism(s) in the abstract and conclusions.

Figure 6 should be cleaned up a bit. Please explain in the legend that the red dots indicate assimilation, and separate Fe2+ and O2 in the top x axis.

p 16 line 22 Some review of known assimilation efficiencies of aerobic and anaerobic methanotrophs would be useful here.

NOx is usually used to denote NO + NO2, and its use in this manustrict to denote nitrate + nitrate will be confusing to many.

---

## Referee Comment (RC3) · Anonymous Referee #3 · 15 Apr 2016

The article of Sturm et al. describes the investigation of methane oxidation pathways in Lake Matano, Indonesia. This unique ecosystem is considered to represent conditions like they existed in Precambrian oceans, and thus understanding of methane fluxes in such an ecosystem would provide valuable information of Earth' early methane cycle. The authors performed incubation experiments with radiolabeled methane and obtained methane oxidation/methane assimilation rates for various depths ranging from oxic to fully anoxic.

General comments: The authors state that methane oxidation in the anoxic water column is supported by oxidized metals or nitrogen oxides, however, the evidence provided to make this conclusions is not sufficient to make these statements. Conclusions are based on theoretical ∆G calculations based on in-situ concentrations of above mentioned potential electron acceptors including sulfate. In my opinion, these calculations can only remotely reflect the actual situation since the fluxes are not considered here. Sulfate concentrations were discussed to be insufficient to explain the observed rates, which led to the conclusion of its minor contribution to AOM. Lake Matano ecosystem has been described as rich in iron and manganese oxides, which could fuel a cryptic sulfur cycle in this lake. Thus, sulfate produced via such a process could potentially still fuel the sulfate dependent AOM without a measurable sulfate accumulation. Incubation experiments of lake water amended with the discussed potential electron acceptors would possibly add more information about their stimulation of methane oxidation rates. Another very valuable addition to unravelling the methane oxidation pathways in different depth intervals would be the investigation of microbial community including 16S rRNA phylogeny and known functional genes. The authors speculate of the involvement of nitrogen oxides in AOM and since the functional genes are known for both nitrite- and nitrate dependent AOM, it would be interesting to see whether at least the so far known organisms are involved. Moreover, the authors show that the substantial amount of methane must be assimilated into biomass. I wonder what part of the microbial community id responsible for the calculated methane uptake.

In general, the authors should either weaken their conclusions about the involvement of alternative electron acceptors in observed methane oxidation or provide additional evidence to support the current statements.

Specific comments to introduction: Lines 20-23: M. oxyfera enrichment cultures were shown to perform nitrite-dependent anaerobic methane oxidation. Please correct the text. The enrichment culture of archaea performing nitrate-dependent methane oxidation was described to produce nitrite (not ammonium) as the main end product which provided substrate for anammox bacteria (Haroon et al., 2013). Please correct the text.